



# Consistent EO Land Surface Products including Uncertainty Estimates

Thomas Kaminski[1], Bernard Pinty[2], Michael Voßbeck[1], Maciej Lopatka[3], Nadine Gobron[2], and Monica Robustelli[2]

[1]The Inversion Lab, Tewessteg 4, 20249 Hamburg, Germany
[2] European Commission, Joint Research Centre (JRC), Directorate for Sustainable Resources, Knowledge for Sustainable Development and Food Security Unit, TP440 Via E. Fermi 2749, I-21027 Ispra (Va), Italy
[3]European Commission, Research Executive Agency, Future and Emerging Technologies: FET-Open COV2 17/120, 16 Place Rogier, B-1049 Brussels, Belgium

*Correspondence to:* Thomas Kaminski (Thomas.Kaminski@Inversion-Lab.com)

**Abstract.**

Earth Observation (EO) land products have been demonstrated to provide a constraint on the terrestrial carbon cycle that is complementary to the record of atmospheric carbon dioxide. We present the Joint Research Centre Two-stream Inversion Package (JRC-TIP) for retrieval of variables char-

acterising the state of the vegetation-soil system. The system provides a set of land surface variables that satisfy all requirements for assimilation into the land component of climate and numerical weather prediction models. Being based on a one dimensional representation of the radiative transfer within the canopy-soil system such as those used in the land surface components of advanced global models, the JRC-TIP products are not only physically consistent internally, but also achieve a high

degree of consistency with these global models. Furthermore, the products are provided with full uncertainty information. We describe how these uncertainties are derived in a fully traceable manner without any hidden assumptions from the input observations, which are typically broadband white sky albedo products. Our discussion of the product uncertainty ranges, including the uncertainty reduction, highlights the central role of the leaf area index which describes the density of the canopy.

We explain the generation of products aggregated to coarser spatial resolution than that of the native albedo input and describe various approaches to validation of JRC-TIP products, including the comparison against in-situ observations. We present a JRC-TIP processing system that satisfies all operational requirements and explain how it delivers stable climate data records. As many aspects of JRC-TIP are generic the package can serve as an example of a state-of-the-art system for retrieval

of EO products, and this contribution can help the user to understand advantages and limitations of such products.





## 1 Introduction

This special issue addresses the consistent assimilation of multiple data streams into biogeochemical models. Among the available data streams, long-term high precision observations of the atmospheric carbon dioxide concentration (see, e.g., Houweling et al., 2012) provide an indispensable constraint for the (process parameter) calibration of terrestrial biosphere models in Carbon Cycle Data Assimilation Systems (CCDAS, Rayner et al., 2005). The strength of this constraint is quantified by significant reductions of uncertainty in simulated terrestrial carbon fluxes diagnosed over (Kaminski et al., 2002; Rayner et al., 2005) or predicted after (Scholze et al., 2007; Rayner et al., 2011) the assimilation window. In recent multi-data stream assimilation studies at global scale (Scholze et al., 2016; Schürmann et al., 2016) the constraint through the flask sampling network has proven essential to achieve realistic magnitudes of the terrestrial carbon sink. The flask sampling network alone does, however, only constrain a sub-space of the space of unknown process parameters. Thus, additional, complementary, constraints are required to further reduce uncertainties in the system. Such complementarity has been demonstrated for Earth Observation (EO) products (Gobron et al., 2007; Pinty et al., 2011b) of the Fraction of Absorbed Photosynthetically Active Radiation (FAPAR), which provide information on, e.g., the vegetation phenology and colour. The effect on carbon and water fluxes of assimilating FAPAR in addition to atmospheric carbon dioxide samples is, for example, quantified by Kaminski et al. (2012) and Schürmann et al. (2016).

The assimilation of an EO product such as FAPAR requires the capability to simulate (by a so-called *observation operator*) its counterpart from the model's prognostic variables, i.e. the variables that the integration scheme of the model's dynamical equations steps forward in time (Kaminski and Mathieu, 2016). For a land product such as FAPAR, the construction of the observation operator requires to solve the equations for the radiative transfer (RT) within the canopy-soil system. The RT within the canopy is complicated as the leaves, which scatter the solar radiation, are large (compared to the wavelength) and vary in their orientation and optical properties. For large-scale terrestrial models it is (at least computationally) infeasible to resolve the small-scale three-dimensional heterogeneity of the canopy. The most advanced RT representations in such models are one-dimensional approximations relying on so-called two-stream or (two-flux) approaches.

The retrieval of a set of EO products describing the evolution of the canopy-soil system, e.g. leaf area index (LAI) or FAPAR, also has to rely on a RT model, in EO terminology called *forward model*, to simulate the partitioning of the incoming solar radiation into contributions from the individual radiative fluxes, i.e. those absorbed in, transmitted trough, and reflected by the canopy. In order to exploit the full potential of EO, this forward model should be as close as possible to the RT model used in the observation operator for assimilation. The joint retrieval of a set of EO products with the same RT model is a pre-requisite to assure physical consistency (including conservation of energy) of the retrieved products. The use in a CCDAS requires the retrieval product to be provided with a (typically space- and time-dependent) uncertainty estimate. For assimilation of multiple products



from a joint retrieval the correlation of their uncertainty is also required to allow the extraction of
the true information content from the jointly retrieved products. The retrieved products must be
quality assured, i.e. they need to be validated against independent information. Finally, the retrieval
algorithm must be efficient enough to allow global-scale processing, preferably near real time.

The Joint Research Centre Two-stream Inversion Package (JRC-TIP, Pinty et al., 2007, 2008) is
a retrieval package that fulfils the above conditions. It is built around a two-stream model (Pinty
et al., 2006) of the RT in the canopy soil system (see section 2) and applies a joint inversion (Taran-
tola, 2005) approach (see section 3) that combines the information in observed radiative fluxes with
prior information on the model parameters (see section 4.1). Its products are posterior estimates of
the model parameters, i.e. effective LAI, spectrally variant background reflectance, effective canopy
reflectance and transmittance (where effective indicates model-dependence (see section 2) and all
radiant fluxes, including (but not limited to) model counterparts to the ones that have been observed.
The retrieved products are available with uncertainty estimates and their covariance (sometimes
termed error covariance). The package is highly flexible: It can be operated for any combination of
narrowband, broadband, or hyperspectral radiation flux observations (Lavergne et al., 2006) and on
all spatial scales above 100 m (when lateral flux components can safely be neglected) even for heigh
canopies. The radiative flux that is accessible to observations from space is the reflected sunlight,
i.e. the albedo, once a complex series of procedures to remove atmospheric effects has been ap-
plied together with performing the required integration over exiting and/or Sun illumination angles.
Hence, for EO applications JRC-TIP is typically set up to use observed albedo as input. Healthy
green vegetation is characterised by a strong albedo difference between the visible (VIS) and near
infrared (NIR) domains of the spectrum. Accordingly, the system is typically operated on albedo
input in these two wavebands. In this configuration it has been applied to broadband albedos derived
from MODIS (Pinty et al., 2007, 2008, 2011a, b), MISR (Pinty et al., 2007, 2008), and Globalbedo
(Disney et al., 2016). Section 4 describes enhancements of robustness and efficiency through the use
of so-called TIP tables, i.e. look up tables of quality-controlled retrievals over a fine discretisation
of the input space (Clerici et al., 2010; Voßbeck et al., 2010). Section 4 discusses products from
a large-scale processing exercise (Pinty et al., 2011a, b) based on MODIS collection 5 broadband
albedo input, with a focus on the reported uncertainty estimates. Validation of JRC-TIP products is
described in section 5.

## 2  Radiative Transfer Model

The two-stream model at the core of JRC-TIP is described in full detail by Pinty et al. (2006). We,
hence, restrict ourselves to a brief summary of the main features. The model is designed to solve
the radiation balance for the canopy-soil system (see figure 1). It simulates the solar radiant fluxes
scattered by, transmitted through and absorbed in a vegetation canopy that is composed of so-called





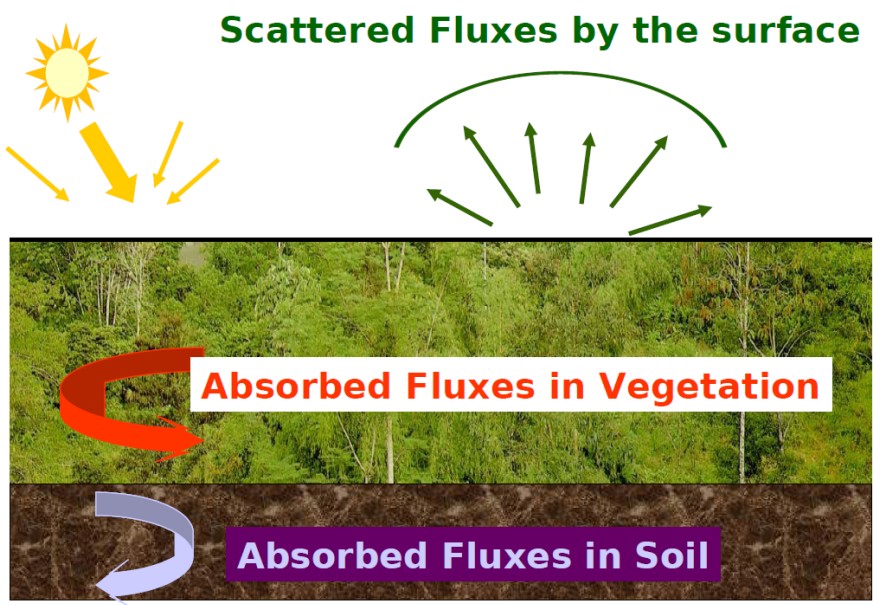

Figure 1: Schematic partitioning of the incoming solar radiation in the canopy-soil system.

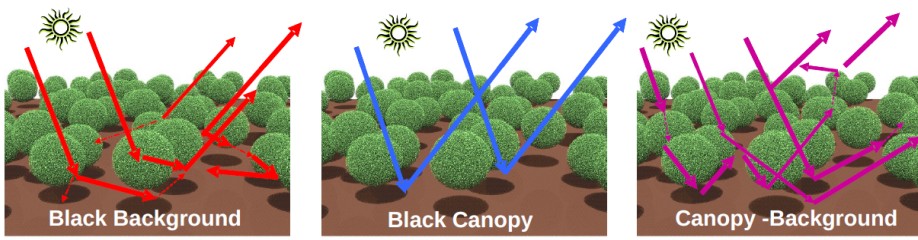

Figure 2: Decomposition of the total flux into three contributing fractions. The two-stream solution applies to the Black background contribution (left hand side).

bi-Lambertian leaves, possibly exhibiting a preferred orientation. The bi-Lambertian leaf scattering
property is such that the fraction of radiation that is not absorbed is scattered as a cosine distribution around the leaf normal vectors. The top and bottom boundary conditions are specified by the downwelling of direct and diffuse radiant fluxes and the albedo of the background, respectively. This model is constructed from dedicated solutions to three separate problems involving 1) the scattering by the vegetation layer only, identified as the black-background contribution, 2) the flux transmitted
directly through the vegetation layer involving only the background, that is the black-canopy contribution and finally 3) the contribution to the upward and downward scattered and transmitted fluxes





involving multiple interactions between both the vegetation layer and its underlying background (see figure 2).

This 1-D model provides a solution to the black background problem which follows the two-stream formulation established originally by Meador and Weaver (1980). It ensures the correct balance between the scattered, transmitted and absorbed radiant fluxes not only for structurally homogeneous but also for heterogeneous canopies. The applicability to heterogeneous canopies relies on the finding by (Pinty et al., 2004, section 3.3) that a solution to a 3-D flux problem satisfying the conditions imposed by a "radiatively independent volume" can always be achieved using a 1-D representation. The model's canopy state variables required for the correct flux representation are, however, so-called effective variables. They deviate from the true canopy variables and are thus only meaningful in the context of this model.

These effective variables are, a spectrally invariant quantity, namely the Leaf Area Index (LAI) and, spectrally dependent parameters that are the leaf single scattering albedo wl = rl + tl and the ratio dl = rl/tl (identified here as the asymmetry factor) where rl and tl correspond to the leaf reflectance and transmittance, respectively. The albedo of the background, rg, is itself defined as the true (by contrast to effective) value and retrieved as such. And clearly, for all fluxes true values are simulated.

The possibility to use a (1D) two-stream representation to solve a flux problem irrespective of the 3D complexity of the scene conditions means that the model can be operated in inverse mode to retrieve a set of state variables for the canopy-soil system that allows an accurate flux representation. The model is implemented in numerically efficient, modular, and portable form, to simplify its integration into climate and numerical weather prediction (NWP) models.

## 3 Inverse Model

JRC-TIP applies the joint inversion approach of Tarantola (2005) (discussed as Bayesian inversion by Rayner et al. (2016) in this special issue): It estimates the state vector (in the following also called parameter vector) from a given set of observations and the available prior information. The a priori state of information is quantified by a probability density function (PDF) in parameter space, the observational information by a PDF in observation space, and the information from the model by a PDF in the joint space, i.e. the Cartesian product of parameter and observation spaces. The inversion combines all three sources of information and yields a posterior PDF in the joint space.

Prior and observational PDFs are difficult to specify. We use Gaussian shapes with respective mean values denoted by $x_0$ and $d$ and respective covariance matrices denoted by $C(x_0)$ (prior parameter uncertainty) and $C(d)$ (data uncertainty). The data uncertainty is the sum of uncertainties





due to errors in the observational process, $C(d_{\mathrm{obs}})$ and errors in our ability to correctly model the observations, $C(d_{\mathrm{mod}})$:

$$C(d) = C(d_{\mathrm{obs}}) + C(d_{\mathrm{mod}}) \tag{1}$$

Some observational products provide uncertainty ranges and their correlation, i.e. the entire $C(d_{\mathrm{obs}})$. If this is not the case, we often assume uncorrelated uncertainties, i.e. zero off-diagonal elements. The diagonals are populated with the squares (i.e. variances) of the 1-sigma uncertainty ranges, for which we typically proceed as follows: In $C(d_{\mathrm{obs}})$ we often use values proportional to $d$ with a floor value. As the value in $C(d_{\mathrm{obs}})$ typically considerably exceeds that in $C(d_{\mathrm{mod}})$ (see section 5) we neglect the latter. The exception is for small values of $d$, where the floor value is supposed to represent $C(d_{\mathrm{mod}})$. Note that, in the typical setup, with $d$ being broadband albedo products, there is no additional contribution from representation error (see, e.g., Heimann and Kaminski, 1999; Kaminski et al., 2010), as the model and the observations are defined on the same space-time grid.

For later use it is convenient to have two separate notations for the model simulation of a flux vector from a given state vector $x$. For simulation of the full vector of all flux components $y$ we use $N$ and when the flux vector is restricted to those components for which we have observations $y_{\mathrm{obs}}$ we use $M$, i.e.

$$y_{\mathrm{obs}} \;=\; M(x) \text{ or} \tag{2}$$
$$y \;=\; N(x) \ . \tag{3}$$

The inverse model is flexible with respect to the number and width of spectral bands that are simulated and the subset of simulated fluxes $y_{\mathrm{obs}}$ that are observed. Every combination is feasible; Lavergne et al. (2006) provide examples.

Since the model is only weakly non-linear, we can approximate the posterior PDF by a Gaussian PDF. The corresponding marginal PDF in parameter space is thus also Gaussian, with mean value $x$ and covariance $C(x)^{-1}$. The mean $x$ is approximated by the maximum likelihood point, i.e. the minimum of the misfit function:

$$J(x) = \frac{1}{2}[(M(x) - d)^T C(d)^{-1}(M(x) - d) + (x - x_0)^T C(x_0)^{-1}(x - x_0)] \tag{4}$$

$C(x)$ is approximated by the inverse of the misfit function's Hessian, $H$, evaluated at $x$:

$$C(x) \approx H(x)^{-1} \ . \tag{5}$$

To understand this relation it is instructive to look at the case of a linear model (denoted by $M'$):

$$H(x) = {M'}^T C(d)^{-1} M' + C(x_0)^{-1} \ . \tag{6}$$



The Hessian is the sum of two terms, one reflecting the strength of the constraint by the prior
information, and the other reflecting the observational constraint. Typically adding the observational
constraint increases the curvature of the cost function which via equation 6 translates to a reduction in
uncertainty compared to the prior. One of the uncommon counter-examples is provided by (Lavergne
et al., 2006).

From the optimal parameter set we can simulate (see equation 3) all radiant fluxes (including the
non-observed ones). To assess the strength of the observational constraint on a simulated radiant
flux, we use $N'$, the first derivative of $n$ to propagate the posterior parameter uncertainties forward
the uncertainty in simulated vector of radiant fluxes $C(y)$:

$$C(y) = N'C(x)N'^{T} \tag{7}$$

Equation 7 is particularly useful for comparing the TIP results with independent observations.

Evaluating 7 for the prior uncertainty $\mathbf{C}(x_0)$ instead of the posterior uncertainty $\mathbf{C}(x)$, i.e. for a
case without observational constraint, yields a prior uncertainty for the flux:

$$C(y_0) = N'C(x_0)N'^{T} \tag{8}$$

For any component of the flux vector we can quantify the added value/impact of the observations
by the uncertainty reduction or knowledge gain relative to the prior.

$$\frac{\sigma(y_{i,0}) - \sigma(y_i)}{\sigma(y_{i,0})} = 1 - \frac{\sigma(y_i)}{\sigma(y_{i,0})}, \tag{9}$$

where, $\sigma(y_i)$ and $\sigma(y_{i,0})$ respectively denote the 1 sigma uncertainty ranges, the squares of which
populate the diagonals of $C(y)$ and $C(y_0)$. For example, if $\sigma(y)_i$ is 90 % of $\sigma(y_{i,0})$, then the uncer-
tainty reduction is 10%; i.e. we have increased our knowledge on $y$ by 10 %.

The simultanous retrieval of all state variables and the associated fluxes within a single model
assures physical consistency between the derived products. This includes simulated counterparts
$y_{\mathrm{obs}}$ of the observed flux components.

This inversion approach is relatively generic, i.e. it similarly applies to further RT models in the
optical domain (see, e.g., Lavergne et al., 2007; Lewis et al., 2012) or other spectral domains, e.g.
the passive microwave domain (see also Kaminski and Mathieu, 2016).

Equation 4 is minimised by a so-called gradient algorithm that relies on code for evaluation of $J$
and its gradient. Further derivative code is used to evaluate equations 5 and 7.





Table 1: Mean values $\mathbf{x_0}$ and associated standard deviations $\sigma(\mathbf{x_0})$ used to set the diagonal of the prior uncertainty covariance matrix $\mathbf{C}(\mathbf{x_0})$. $\Delta_{\lambda_1}$ and $\Delta_{\lambda_2}$ correspond to the broadband visible (0.3–0.7 $\mu$m) and near-infrared (0.7–3.0 $\mu$m) spectral domains, respectively. $\omega_l(\Delta_{\lambda_{1,2}})$, $d_l(\Delta_{\lambda_{1,2}})$ and $r_g(\Delta_{\lambda_{1,2}})$ refer to the effective canopy single scattering albedo, asymmetry factor and background albedo, respectively.

| Variable identification | $\mathbf{x}_0$ | $\sigma(\mathbf{x_0})$ |
| --- | --- | --- |
| LAI | 1.5000 | 5.0 |
| $\omega_l(\Delta_{\lambda_1})$ | 0.1700 and 0.1300 [a] | 0.1200 and 0.0140 [a] |
| $d_l(\Delta_{\lambda_1})$ | 1.0000 | 0.7000 |
| $r_g(\Delta_{\lambda_1})$ | 0.1000 [b] and 0.50 [c] | 0.0959 [b] and 0.346 [c] |
| $\omega_l(\Delta_{\lambda_2})$ | 0.7000 and 0.7700 [a] | 0.1500 and 0.0140 [a] |
| $d_l(\Delta_{\lambda_2})$ | 2.0000 | 1.5000 |
| $r_g(\Delta_{\lambda_2})$ | 0.1800 [b] and 0.350 [c] | 0.2000 [b] and 0.25 [c] |

[a] Value associated with the 'green' leaf scenario.

[b] Value adopted for the bare soil case with a correlation between the two spectral domains of 0.8862 set in $\mathbf{C}(\mathbf{x_0})$.

[c] Value adopted under occurrence of snow with a correlation between the two spectral domains of 0.8670 set in $\mathbf{C}(\mathbf{x_0})$.

## 4 Operational Processing

### 4.1 Prior Information

The radiative flux component that is accessible to observations from space is the reflected flux (albedo). As photosynthesis is driven by absorption in the VIS, our focus is on the flux partitioning in this domain of the spectrum. Pinty et al. (2009) demonstrate that under typical, non-snow conditions and with known optical properties at leaf-level the background reflectance largely determines the albedo in the VIS, and the effective LAI the albedo in the NIR. Hence, it is favourable to operate JRC-TIP in both VIS and NIR, with albedo observations in these two broad bands. Including the NIR brings in one additional observational constraint but also adds three spectrally variant state variables to the inverse problem. This is partly compensated by (approximately) known relations of the background reflectance across the VIS and NIR domains. This relation translates to our inversion formalism as a correlated uncertainty and is visualised by the ellipsoids (indicating the 1.5 sigma uncertainty ranges) shown in figure 3. We note that this relation, known as *the soil line* (see,



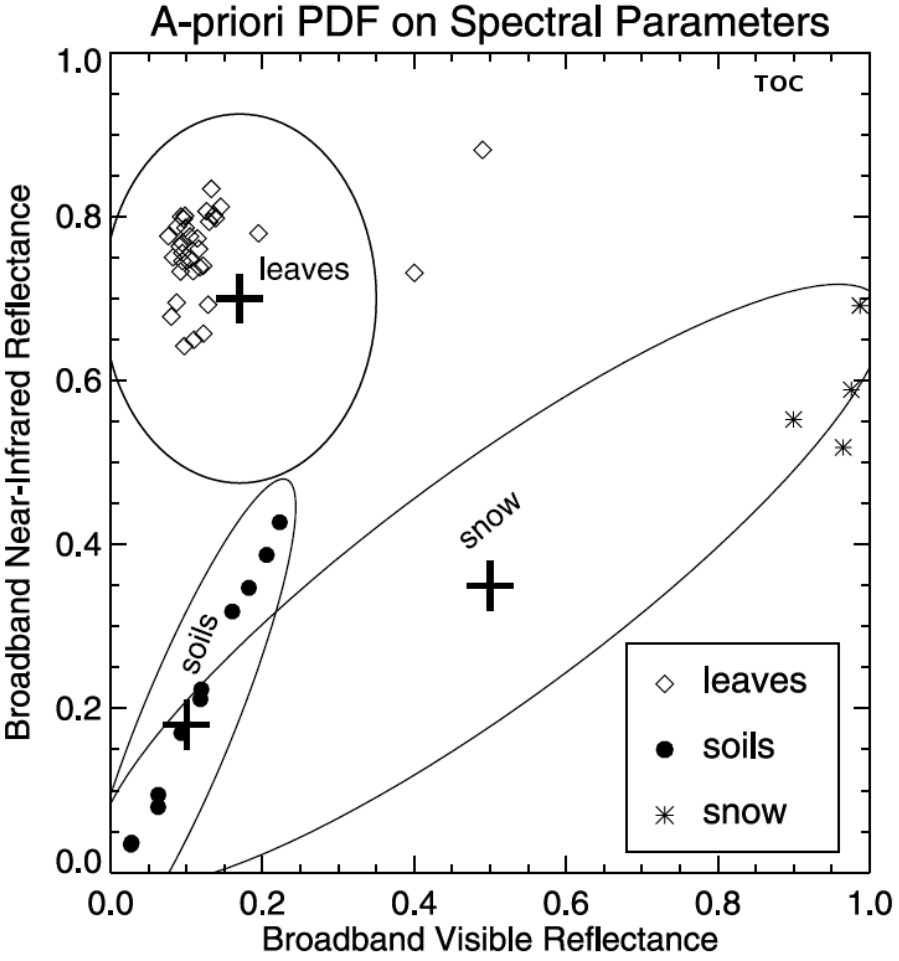

Figure 3: Prior 1,5 sigma uncertainty for effective canopy single scattering albedo and background reflectance. Mofified version of Figure 1 from Pinty et al. (2008).

e.g. Chi (2003)), changes with the occurrence of snow. Figure 3 further shows the components of the prior PDF describing the effective canopy single scattering albedo, which is based on a combination of modelled and observed leaf optical properties (see Pinty et al., 2008, for details). This ellipsoid describes our standard prior, denoted as polyochromatic leaf scenario. We also operate the system with an alternative prior, denoted as green leaf scenario, which is characterised by a much lower un-

certainty and a slight shift in the mean value. Both scenarios are operated with snow and non-snow priors for the background reflectance, depending on additional snow information. A summary of the





prior information is provided in table 1. We stress the high prior variance of 25 for the effective LAI, a deliberately conservative assumption that results in a low weight on the prior term in equation (4).

### 4.2 Observations

The specific quantities to be discussed in section 4.4 have been retrieved (as described by Pinty et al. (2011a)) from the MODIS collection V005 (MCD43B3) broadband white sky albedo (WSA) products at 1 km resolution (Schaaf et al., 2002). The WSA product uses a synthesis period of 16 days, in which observed reflectances under various illumination angles are used to calculate the spherical integral (isotropic illumination) . The albedo product provides a data set every 8 days such
that filtering out every second data set yields a sequence of data sets, in which each member is based on its individual 16 day synthesis period. This procedure maximises the temporal independence of the observational input for JRC-TIP. The MODIS collection V005 WSA product provides a quality flag associated with all spectral bands, but no covariance of uncertainty. As described in section 3 we populate the non-diagonal elements of $C(d)$ with 0. For the diagonal elements the quality flag
such that good (other) quality is mapped onto a one sigma uncertainty range of 5% (7%) relative to the flux and a floor value is set of $2.5 \ 10^{-3}$. All other observations are discarded. In addition the MODIS snow indicator is used to trigger a swich of the prior for the background reflectance from the non-snow to the snow version.

### 4.3 Robustness and Efficiency

In the above-described setup all observations are restricted to broadband VIS/NIR albedo pairs, which can theoretically take values in the two dimensional domain $[0,1]x[0,1]$ (albedo plane). Observations retained for processing with JRC-TIP fall either in the 5% or the 7% uncertainty case. For both uncertainty cases we now apply JRC-TIP over a discretisation the albedo plane with step size of $10^{-3}$ (i.e. a factor of 2.5 below the minimum uncertainty) on both axes (Clerici et al., 2010;
Voßbeck et al., 2010). This provides us with a set of $2 \ x \ 10^{-3} \ x \ 10^{-3} = 2$ million JRC-TIP retrievals which populate the 2D space of theoretically possible albedo input observations that is dense enough for all practical purposes. We note that, in practice, only a subdomain of the albedo plane is covered by observations. figure 4 shows all the location in the albedo plane of all albedo pairs used by (Pinty et al., 2011a) for their processing of the year 2005, excluding those with snow-flag. Switching
from a non snow to a snow prior adds a factor of 2, i.e. there are in total 4 million retrievals for the polychromatic leaf scenario.

We denote the above-described set of retrievals as the TIP table. Once the TIP table is generated, a retrieval for any given albedo input pair can be performed through a look up in the TIP table. We stress that the role of the TIP table is different from the traditional use of look up tables (LUTs)
in retrieval schemes: While traditional LUTs relate input to output of the *forward model* (i.e. state variables to albedos), the TIP table relates input to output of the *inverse model* JRC-TIP (i.e. albedos



to the complete set of variables retrieved by TIP, including uncertainty ranges and auxillary information). The use of the TIP table in a processing system has four advantages over the use of a standard JRC-TIP retrieval.

1. First, it is faster. For example, processing of one period globally in 500m spatial resolution takes less than 2 hours in a single core of a standard desktop computer equipped with an Intel Core-i5 CPU running at a maximum frequency of 3.3GHz. The value of 2 hours covers the full I/O (including generation of output files) and yields an average processing time per pixel of less than 0.004ms.

2. Second, it simplifies quality control: Clerici et al. (2010); Voßbeck et al. (2010) describe a number of iterative procedures to enhance the quality of the retrievals in TIP table. They exploit, for example, the requirement of a smooth dependence of the solution on the input albedos to detect outliers.

        3. Third, the TIP table approach assures stability of any Climate Data Record (CDR) that is gen-
erated from a stable albedo CDR. By construction, JRC-TIP will always retrieve the same values for all variables and uncertainties, from the same albedo input with the same uncertainty range. For the standard JRC-TIP retrieval this is only guaranteed when the computing environment remains unchanged.

        4. Fourth, a processing system relying on the TIP table is agile. When a compoment of the JRC-
TIP retrieval procedure is improved, the only change required in the processing system, is an update of the TIP table.

For albedo input products that provide per-pixel uncertainty ranges, the TIP table uses a finer discretisation and further dimensions in the uncertainty space, but the same general approach applies. For example, Disney et al. (2016) use a two-dimensional uncertainty space: One dimension each for
uncertainties in VIS and NIR, an extra dimension for their correlation was not included.





### 4.4 Understanding Uncertainty

We analyse the JRC-TIP products over the range of the albedo input plane that is actually covered by observations, more specifically the range covered by the MODIS collection 5 albedo 1 km input products for the year 2005 that were processed by Pinty et al. (2011a, b). We focus on snow-free

background conditions, i.e. all prior values and uncertainties are spatially invariant. We show for effective LAI and background reflectance in the VIS (figure 4) as well as for effective canopy single scattering albedo in the VIS and FAPAR (figure 5) the retrieved mean values (top panels) and one sigma uncertainty ranges (middle panels) as well as uncertainty reduction/knowledge gain as defined by equation (9) (bottom panels) over the albedo plane. The first point to note is the limited sub-set

of the albedo plane that is covered by actual albedo observations. A further point to highlight is the fundamental role of the effective LAI: High effective LAI values correspond to relatively high posterior LAI uncertainty and little knowledge gain, because the dense canopy can only be penetrated to a limited extent. For the same reason, we can infer little information on the background under dense canopies, i.e. there is a high posterior uncertainty and little knowledge gain. By contrast, given the

large amount of canopy material, we can substantially reduce the uncertainty in the single scattering albedo, i.e. we have a large knowledge gain. Low effective LAI characterise an almost transparent canopy: Uncertainty on LAI and background reflectance is low and there is high knowledge gain from observations. The low amount of canopy material limits the knowledge gain for the single scattering albedo, i.e. we are left with relatively high uncertainty. In this regime the observed albedo is

determined by the background reflectance (shown for the visible domain in panel b of figure 4). The pattern of the mean value for FAPAR is similar to that for LAI. The uncertainty is, however, different. While the LAI uncertainty grows steadily with LAI itself, the FAPAR uncertainty exhibits two separated domains with high uncertainty. On the line of constant WSA NIR, one peak is located at WSA VIS around 0.03 and the other peak around 0.13. As pointed out by Pinty et al. (2011b), this reflects

the influence of the soil background, which for LAI values in the range from 0.3 to 0.5 is exhibits an equally complex uncertainty structure (panel d of figure 4). In the minimum of the misfit function $J$ of equation (4) that is displayed in the bottom right panel of figure 5 we can clearly distinguish the two regimes. In the regime of low LAI (located on the soil line) the observations are primarily fit by variation of the background reflectance, and the misfit primarily reflects the deviation from the prior

background reflectance shown in figure 3. Owing to the large prior uncertainty of 5 the contribution of the effective LAI to the prior term of the misfit function is very small, and the remaining canopy variables remain close to the prior. By contrast, in the regime of high LAI the observations are primarily fit through variation of the canopy variables, and the background reflectance is close to the prior.







Figure 4: Mean value (upper panel), Uncertainty (middle panel), and Uncertainy reduction (bottom panel) for effective LAI (left) and background reflectance in the VIS (right)





Figure 5: Mean value (upper panel), Uncertainty (middle panel) for effective canopy single scattering albedo (left) and FAPAR (right). Uncertainy reduction for effective canopy single scattering albedo (bottom left) and misfit function (see equation (4)) at minimum (bottom right).





### 4.5 Aggregation

For data assimilation at global scale, products are required in lower spatial resolution than the native resolution of the albedo input product (which currently is in the order of 1 km). JRC-TIP retrieval and aggregation are not commutative, i.e. it makes a difference whether we aggregate JRC-TIP products generated at the native resolution of the albedo product or operate JRC-TIP on the aggregated albedo input. This is illustrated on figure 6, which shows retrieved LAI and FAPAR over the albedo plane (with 5% uncertainty) and indicates two albedo pairs by black points. If we assume the two albedo pairs describe observations over neighbouring pixels of the same area, running JRC-TIP on the average albedo yields LAI and FAPAR values of 0.194 and 0.153. By contrast, the average LAI and FAPAR for both pixels are 0.532 and 0.275, respectively.

Also, when aggregating at the level of JRC-TIP products it is not guaranteed that the aggregated state variables are still consistent with the aggregated fluxes. Only by operating the JRC-TIP on the aggregated albedo input, we can ensure the physical consistency among the JRC-TIP products and with the one dimensional representation of the radiation transfer process in the climate or NWP model. The effect of changing the order of aggregation is quantified by figure 7 which shows on global scale and for a particular period in 2005 the relative differences of FAPAR products derived by either aggregation order. Another point is that the aggregation also needs to be performed on the uncertainty. This requires a specification of spatial uncertainty correlation and is certainly less complicated at the albedo level than at the level to JRC-TIP products.

## 5 Validation

The validation of the JRC-TIP and its generated products is achieved through a variety of complementary stages. The first one consists in assessing the performance of the direct model, namely the two-stream model that is further used in inverse mode to generate the JRC-TIP products. This performance can be thoroughly benchmarked against comprehensive 3D Monte-Carlo models for a series of virtual canopies exhibiting different levels of complexity regarding the radiation transfer regime that these canopies can represent (see section 3 of Pinty et al., 2006). The RAdiation transfer Model Intercomparison (RAMI) initiative (http://rami-benchmark.jrc.ec.europa.eu) offers such a platform for a range of simple and very complex canopy scenarios (Pinty et al., 2001; Widlowski et al., 2007). The 1D model implemented in JRC-TIP was found to be in very good agreement, i.e., better than 3% in most cases, with albedos from accurate and realistic simulations of complex 3D scenarios in both the red and near-infrared spectral regions.

While this first set of RAMI excercises addressed the accuracy of simulated albedo, i.e. $C(d_{\mathrm{mod}})$ in equation (1), a further exercise in the RAMI frame (termed RAMI4PILPS) addressed the accuracy and consistency of the absorbed, reflected, and transmitted radiative fluxes retrieved by inverse models of the soil-vegetation-atmosphere transfer (Widlowski et al., 2011). This exercise thus offers a



possibility to assess the performance of the JRC-TIP with regard to its ability to partition the incom-
ing solar radiation. For the extreme conditions of computer reconstructed 'actual' canopy scenarios,
with a range of sun zenith angle and vegetation background including snow covered conditions, the
vast majority of the absorbed flux values (i.e. FAPAR) falls within +/- 10 % relative to those values
estimated by the reference Monte-Carlo model (see section 3 of Widlowski et al., 2011).

The capability of the JRC-TIP to reconstruct solar fluxes that can be currently measured in-situ
by dedicated instruments, e.g., direct or diffuse canopy albedos and transmission, offers a definite
solution to assess the performance of the procedure. However, the crux of the matter with such an
exercise lies in the large spatial variability of the canopy at various scales such that the spatial and
temporal sampling of a given site must be achieved carefully and quite extensively. A first attempt to

evaluate the JRC-TIP products generated from MODIS white sky albedo input values over a fluxnet
site is described in Pinty et al. (2011c). In this study the authors have capitalized on an ensemble
of LAI-2000 measurements systematically acquired over multiple years along a 400 m transsect as
well as series of photos taken from a tower emerging the top of this deciduous mid-latitude forest.

We note the one to one relation of the direct transmission, $T^{UnColl}$, and effective LAI, $\tilde{LAI}$,

through the Beer-Bouger-Lambert Law

$$T^{UnColl}(\mu_0) = exp(-\frac{1}{2}\frac{\tilde{LAI}(\mu_0)}{\mu_0}), \tag{10}$$

where $\mu_0$ denotes the cosine of sun zenith angle (Pinty et al., 2006, 2009), i.e. $\mu_0 = 1$ when the sun
is at nadir. Figure 8 shows these observations together with the direct transmission derived by JRC-
TIP from MODIS collection 5 broadband WSA products at 500m and 1km. Grey and blue shaded

ranges indicate the spatial variability along the transect at which the observations were collected,
and the red error bars indicate the uncertainty range that is part of the retrieved product. The left-
hand panel is based on 500m MCD43 input albedos and exhibits slightly better fit to the in situ
observed fluxes than the right-hand panel, which is based on the MCD43 1km albedo product. In
this example the root mean squared error (RMSE) is used (see upper right corner in each panel)

as a simple metric that quantifies the fit. Temporal correlation or more sophisticated metrics that
take the uncertainty in products and in observations into account are possible alternatives. We point
out that the uncertainty ranges that are displayed for observed and retrieved transmittance capture
different aspects of uncertainty: While the ranges in the observations cover spatial variability along
the transect, the product error bars refer to the pixel average and indicate the one-sigma uncertainty

range that is consistent with the uncertainties in the prior and in the albedo input.

In general, the results show good consistency between the JRC-TIP products and this ensemble
of information given that the MODIS sub-pixel variability corresponds to a range of values that are
analogous to the uncertainties associated with the JRC-TIP retrievals. For a single period (from mid
to end of January) the direct transmission derived by JRC-TIP from both products is completely

outside the observed range. For the 500m resolution, we trace this back to the input albedos (shown
in figure 9, panel a), where the VIS albedo shows an abnormal increase for this particular period,





which is very likely due to snowy background conditions that remained undetected in the MODIS product, i.e. the snow flag was not raised. The inversion procedure, being operated with non-snow priors in this case, needs to minimise the misfit function $J$ (see equation (4)) which quantifies the

misfit between modelled and observed albedos and the deviation of the parameters from their priors. In order to best fit this high observed albedo in the VIS without being penalised by a high prior term in $J$, the minimisation procedure increases the background reflectance in the VIS (panel (c) in figure 9) and turns off the vegetation contribution by setting LAI close to zero (panel (b) in figure 9), which explains the high direct transmission derived for this period (figure 8), and also means that

there is no absorption of the incoming radiant flux by the vegetation (figure 9, panels (e) and (f)). For the time period in question, the graphs also include, in magenta colour, a second retrieval with snow prior. The corresponding LAI, transmission in the VIS, and the absorption in the VIS and the NIR are then much closer to the values for the preceeding and succeeding periods, and the background reflectance closer to the soil line for snow. We note that our global-scale processing setup scans

non-snow retrievals using several conditions for outliers which may then be corrected by a snow retrieval.

While proposing a simple protocol to validate the JRC-TIP products against in-situ data, Pinty et al. (2011c) also highlighted the lack of critical, although not challenging, measurements of for instance the background albedo and its spatio-temporal variability at site level. This is a typical but

very unfortunate situation, as the combination of the direct transmission (i.e. effective LAI) and the background reflectance, largely determines the partitioning of the incoming flux between the canopy and the soil. It has been so far very challenging to identify other such sites where comparable datasets acquired in-situ over time are available for in-depth validation exercise.

Another element of the validation strategy consists in the comparison of JRC-TIP products derived

over the same location from multiple albedo input products. For example Pinty et al. (2007) analyse differences between JRC-TIP products derived from MODIS and MISR broadband WSAs. Pinty et al. (2008) apply the same strategy but to high- and mid-latitude sites, as their focus lies on the behaviour of JRC-TIP products in the presence of snow. MISR is an instrument flying on the same platform (called Terra) as one of the MODIS instruments, and the procedure for deriving a WSA

comparable to the standard MODIS product is described by Pinty et al. (2007). As MISR observes each pixel from several angles, it can collect a high number of samples and thus provide a good angular integral of reflectance, i.e. a good WSA reference.

Another level of validation is the comparison of JRC-TIP products against products derived with alternative retrieval approaches. An example is presented by Disney et al. (2016), who compare ef-

fective LAI and FAPAR products derived by JRC-TIP with the operational MODIS LAI and FAPAR products (Knyazikhin et al., 1998) at site, regional, and hemispheric scales.

A final level of validation is implicitly performed by the product users in their respective applications. Such applications include analyses of the consistency of the long term CDR, and its inter-



annual variability as demonstrated for FAPAR by Gobron (2015). Sippel et al. (2016) use a deviation
of the 2012 spring and late summer FAPAR from the respective long-term means to analyse the effect
of a drought event on vegetation activity over North America and explain the response mechanism
of the carbon balance as inferred from other data streams (Wolf et al., 2016). A consistency check
against other data streams and a process model is provided by simultaneous assimilation of the FA-
PAR product with further data streams, in particular atmospheric carbon dioxide record (Kaminski
et al., 2013; Schürmann et al., 2016). Consistency to further data streams is also implicitly checked
in diagnostic model setups, for example when the FAPAR product is used as a forcing field for
simulation of photosynthesis (Chevallier et al., 2016).

## 6 Conclusions

The JRC-TIP is a highly flexible retrieval system that delivers a set of radiatively consistent land
surface products. These products include all radiant fluxes (absorbed, transmitted, and reflected)
and the complete set of state variables that parameterise the two-stream model at its core. This
two-stream model provides a one-dimensional approximation of the radiative transfer within the
canopy soil system, typically implemented in advanced land components of climate models. This
renders the retrieved (model dependent) state variables (such as the effective LAI) as compliant as
possible to climate model applications (climate model compliance). The retrieved fluxes have a clear
physical definition and are, thus, model independent. Hence, among the JRC-TIP products the fluxes
are particularly suitable for assimilation into terrestrial models. Even in this case it is, nevertheless,
crucial to have in the terrestrial model an observational operator that provides a correct mapping from
the state variables onto the simulated counterpart of the flux component that is being assimilated.

All JRC-TIP products include estimates of uncertainty including their covariance that are consis-
tently derived in a fully traceable manner through rigorous uncertainty propagation from prior and
observational information in a two-step procedure. The first step derives uncertainty estimates for
the state variables and the second step maps these uncertainty estimates forward to the simulated
fluxes.

For global-scale processing JRC-TIP is operated on broadband albedo products (including snow
information) derived from EO with space and time invariant prior (except in the event of snow) such
that the retrieved products are exclusively based on the EO input. Owing to this low-dimensional
space of observational input an operational system can be set up to retrieve products from a data
base (TIP table) of pre-calculated quality-controlled JRC-TIP solutions (including full uncertainty
quantification). Such a system is computationally extremely efficient, robust, and agile. By construc-
tion it generates temporally stable climate data records from any albedo input record that fulfils this
condition.





JRC-TIP products are typically provided in the native resolution of the albedo input product, i.e. on grids that are much finer (e.g. a few 100 to a few 1000 m) than typical resolutions of continental to
global-scale terrestrial models. To ensure their radiative consistency and climate model compliance, products on grids coarser than this native resolution have to be derived by first aggregating the albedo input and then applying JRC-TIP.

The JRC-TIP methodology is to a large extent generic (see, e.g., Kaminski and Mathieu, 2016) and can be generalised to further RT schemes. This holds in particular for the two-step procedure
that first solves for the state variables and from there then simulates a set of target quantities, both steps including uncertainty propagation. The application of a solution data base requires a bounded, low-dimensional space of observational inputs.





*Acknowledgements.* We acknowledge the support from the International Space Science Institute (ISSI). This publication is an outcome of the ISSI's Working Group on "Carbon Cycle Data Assimilation: How to consis-

tently assimilate multiple data streams". The MODIS products were obtained from the online Data Pool, courtesy of the NASA Land Processes Distributed Active Archive Center (LP DAAC), USGS/Earth Resources Observation and Science (EROS) Center, Sioux Falls, South Dakota, https://lpdaac.usgs.gov/data_access/data_pool.





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

**Data and Code Availability**

C and Fortran implementations of the two-stream code are available at http://fapar.jrc.ec.europa.eu.
The JRC-TIP product based on MODIS collection 5 in 1 km resolution is available upon request to
the authors.





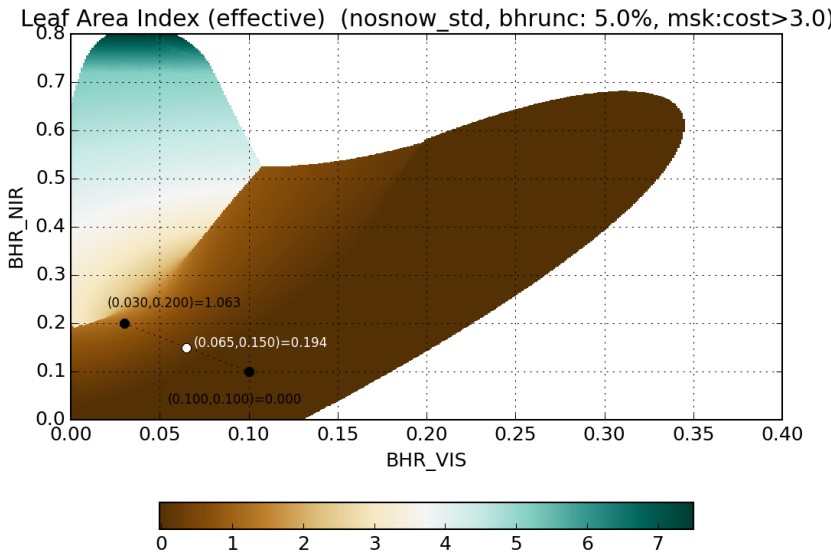

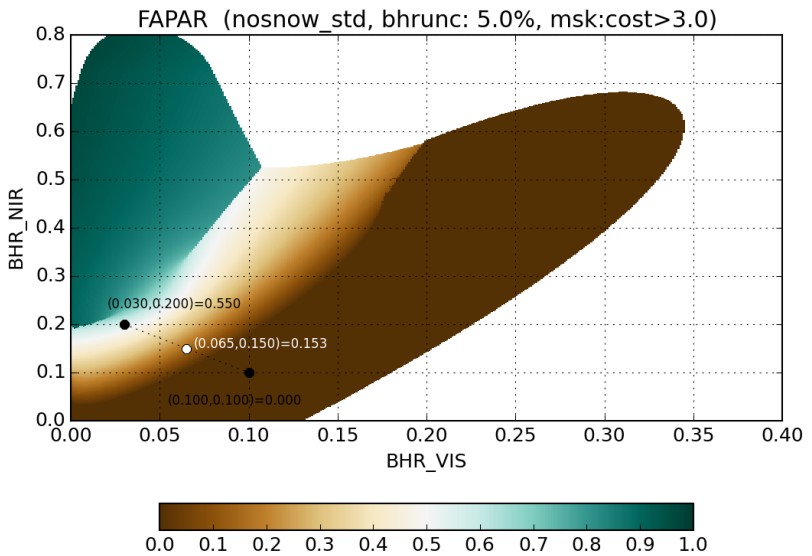

Figure 6: LAI and FAPAR over albedo spectral plane.





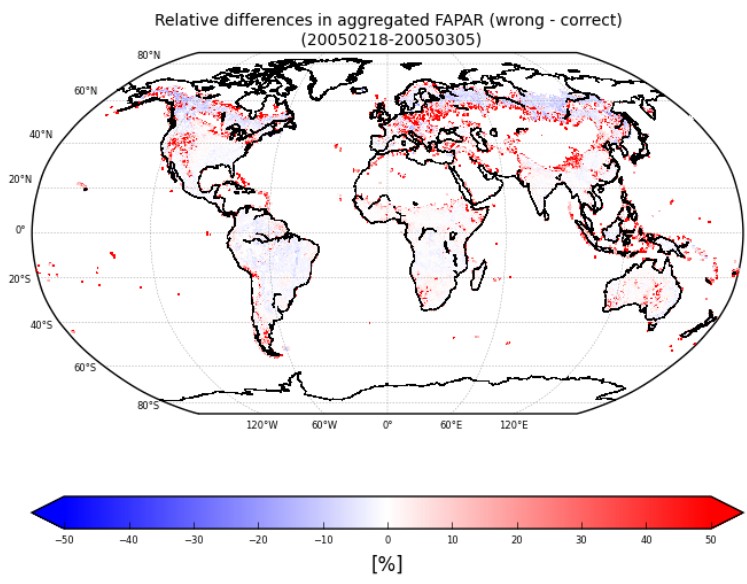

Figure 7: Relative differences of JRC-TIP FAPAR differently aggregated to 0.5°. *wrong* denotes aggregation of the JRC-TIP FAPAR generated at 0.01°, whereas *correct* denotes aggregation of the input albedo products and subsequent application of JRC-TIP.





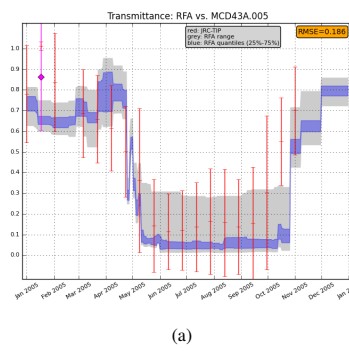

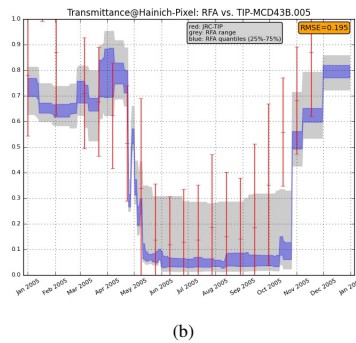

(a)  (b)

Figure 8: Comparison of the fraction of solar radiation transmitted to the background as derived by JRC-TIP applied to MODIS collection 5 input albedos and as observed in situ over the Hainich deciduous forest site following the approach by Pinty et al. (2011c). The left-hand panel is based on 500m MCD43 input albedos and the right-hand panel is based on the MCD43 1km albedo product. The shaded zones indicate the range (grey) and interquartile (blue) range estimated both from the collected in situ measurements (using a LAI-2000 Plant Canopy Analyzer) from year 2002 to 2008 along a 400m transect. Panel a also shows, in magenta colour, a second retrieval for the period from mid to end of January using the snow prior.





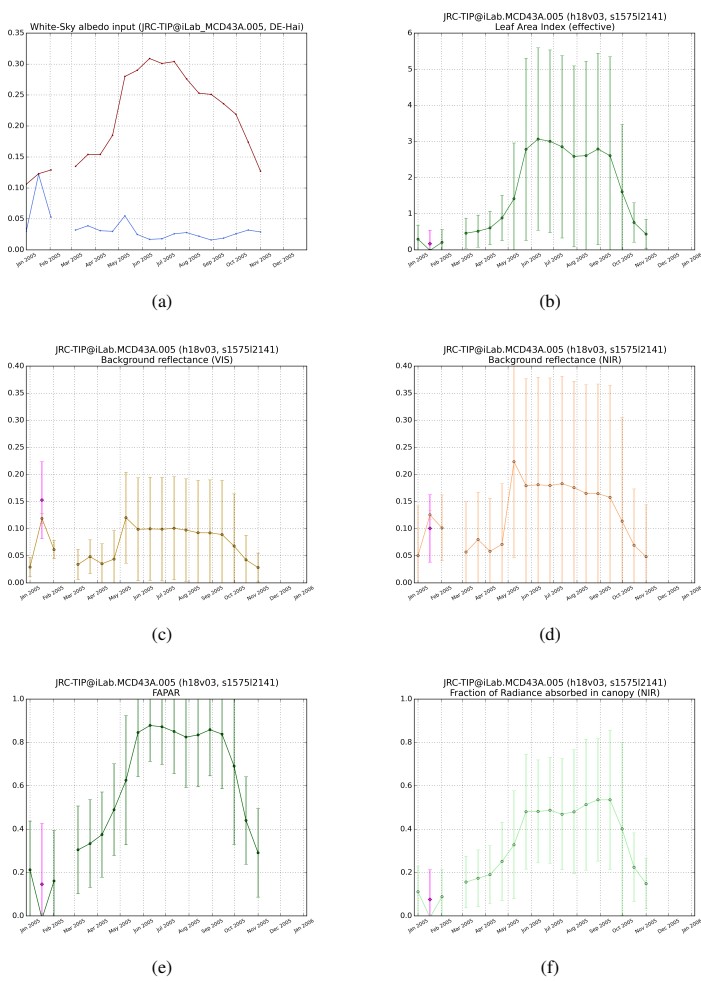

Figure 9: MCD43 500m input albedos over the pixel covering the 400m transect in the visible and near infrared domain (a), retrieved effective LAI (b), background reflectance VIS (c) and NIR (d), fluxes absorbed in the vegetation in VIS (e) and NIR (f). Additional snow prior retrieval in (b)-(f) in magenta.