# Peer review of "Consistent EO Land Surface Products including Uncertainty Estimates"

_Biogeosciences, 2016_

## Referee Comment (RC1) · Anonymous Referee #1 · 24 Oct 2016

The manuscript describes a system to retrieve LAI, FAPAR, and background albedo from visible and near-infrared albedo satellite observations. The work is sound and thus I have no major comments regarding the described retrieval system or the presented results. However, the text is very difficult to read because very often sentences are too long and too complex, technical terms are often introduced without any prior explanation, or topic change within in one sentence. Consequently, the manuscript might be largely incomprehensible to readers with little background in the retrieval of EO products or with no knowledge in inverse modelling. However, the title suggests that the paper describes a generation and evaluation of EO products which will be likely often cited by users of these products. Consequently, the paper should be written so that is comprehensible to a wide user community. In summary, I request to carefully revise the manuscript in terms of language and writing style.

[Figure]

Title of the manuscript

I think that the title of the manuscript does not fit to the content. From the title, I'm expecting a description of the retrieval system and a detailed analysis and evaluation of the products. Furthermore, I'm expecting a link where to receive the data or any other statement about how to receive the data. However, the manuscript mainly describes the retrieval system and some key properties of the retrieved products. The chapter on the product validation is rather a discussion on several possibilities how to validate the system or the products than an intense validation. Therefore I suggest to revise the title of the manuscript so that it rather gives a focus on the description of the processing system than the products.

Writing style

Many sentences are generally difficult to read because #1 they are very long; #2 they cover several topics, concepts, or ideas; or #3 the flow is interrupted by insertions like "e.g.", "i.e", "for example", "however", or by references. Such sentences are for example on lines 27-30, 33-37, 37-39, 40-43, 44-46, 50-53, 63-67, 113-116, 215-217, and 341-344. I suggest to shorten sentences, to remove insertions, and to place references at the end of sentences. Additionally, term like "for example" or "however" should be rather at the beginning of sentences than within sentences. The text needs to be revised in order to reduce the complexity of sentences and to make it more comprehensible.

Detailed remarks

- Lines 28-30: The sentence is difficult to understand because the position of references split the flow. Please place all references at the end of the sentence.

- Lines 40-41: The text requires a high level of knowledge from the reader. For example, in this sentence the term "observation operator" is introduced just within two brackets. I think the text could be much more comprehensive if these kind of insertions are removed from sentences and are explained of defined afterwards in a second

sentence.

- Lines 93-96: This part would be easier to understand if you shortly define Lambertian scattering.

- Line 238: Use capital letters for the beginning of the sentence.

- Lines 294-296: "is exhibits" is likely wrong.

- Line 297: "bottom right panel" = Fig. 5 f

- Line 354-369: The purpose of this paragraph is unclear to me. Please provide first an introductory sentence that describes why the Beer-Bouger-Lambert law is introduced here and how it relates to the validation of the product.

- Figure 7: Please explain shortly potential reasons for the differences because they seem to be associated with certain vegetation types, e.g. overestimation of "wrong" in transitional regions and underestimations in boreal forests.

- Table 1: Can you please indicate in the table legend or footnote why there are two mean values for the $\omega\_l$ parameters.

- Figure 3: The figure would be much easier to understand if you could provide some more details on the shown data without forcing the reader to go to the original publication. What do the points represent? Are these results from measurements or model outputs? From which regions or ecosystems are the measurements?

- Figures 4 and 5: The colour legends show that black is used for snow. However, I'm irritated from this legend because I don't see that black colour is used in any of these plots. Additionally, I think it is not necessary to indicate snow in these plots. Do I overlook something or could you remove the snow/black entry of the legend?

- Figure 8: The figure and the labels are too small. Please increase especially the size of labels.

---

## Referee Comment (RC2) · Anonymous Referee #2 · 24 Nov 2016

General Comments:

The paper presents a sound application of the previously developed Joint Research Center Two-stream Inversion Package (JRC-TIP) to retrieve radiatively consistent land surface products, specifically all radiant fluxes (absorbed, transmitted and reflected) and LAI and FAPAR state variables, that are compliant with climate and numerical prediction models. Full and robust uncertainty estimates are also provided both for the state variables and the fluxes. The authors show that global-scale processing is also achived in an extremely efficient, robust and agile computational way. The JRC-TIP retrieval system also provides aggregated coarser grid products that are better compliant with continental to global-scale terrestrial models. They also describe various approaches to validation of the JRC-TIP products, including the comparison with in-situ observations from dedicated instruments/sites.

Specific comments following the requested aspects:

1. Does the paper address relevant scientific questions within the scope of BG? Yes, the paper shows the suitability of the previously developed JRC-TIP retrieval system to provide land surface parameters, including their respective full and rigorous uncertainty information, ready for assimilation into climate and numerical weather prediction models. The paper demonstrates this for LAI and FAPAR, also providing their lower resolution coarser aggregated products and validation performance, even with in situ dedicated instruments.

2. Does the paper present novel concepts, ideas, tools, or data? Yes, the paper shows the robustness and efficiency of the JRC-TIP package which could also be applied to new Copernicus products.

3. Are substantial conclusions reached? Yes, definitely: • Flexibility of JRC-TIP to deliver radiatively consistent land surface products including their fluxes (absorbed, transmitted and reflected) suitable for assimilation into terrestrial models • Rigurous estimation and fully traceability of the uncertainty both for the state variables and for the simulated fluxes • Computationally efficient, robust and agile global-scale processing • Provision of aggregated coarser grids products better compliant with climate models • Possibility of various approaches to validation, including in-situ observations from dedicated instruments/sites

4. Are the scientific methods and assumptions valid and clearly outlined? Yes. It should be mentioned that the detailed scientific explanation of the inverse modelling approach requires high knowledgdable proficiency on the matter. The excellent scientific basis of the senior co-authors guarantees this. They are absolute reference in the field.

5. Are the results sufficient to support the interpretations and conclusions? Yes. The work is presented for LAI and FAPAR. However, the title is ampler mentioning "Land Surface Products" in general. A minor suggestion is to better specify the title. The work has not been tested for many other different land surface products (soil moisture,

evapotranspiration, etc).

6. Is the description of experiments and calculations sufficiently complete and precise to allow their reproduction by fellow scientists (traceability of results)? Yes, but it is required that fellow scientists trying to reproduce the work should be proficient in the matter, as indicated in #4. Besides, the authors make the C and Fortran versions of the code available upon request.

7. Do the authors give proper credit to related work and clearly indicate their own new/original contribution? Yes. This is recognised in the generous and appropriate reference list, as well as in the Acknowledgment section.

8. Does the title clearly reflect the contents of the paper? The title is more general that the applications actually shown in the paper. As indicated in #5, the title mentions "Land Surface Products" in general. However, the application is only for LAI and FAPAR. It is suggested to be more specific in the title.

9. Does the abstract provide a concise and complete summary? Yes, the abstract is very well and clearly written.

10. Is the overall presentation well structured and clear? Yes.

11. Is the language fluent and precise? Yes, as far as this referee may judge.

12. Are mathematical formulae, symbols, abbreviations, and units correctly defined and used? Yes.

13. Should any parts of the paper (text, formulae, figures, tables) be clarified, reduced, combined, or eliminated? A minor suggestion is to unify the format and style of the figures, namely Figure 3, Figures 4 and 5, Figure 6, and Figures 8 and 9. They are all graphs but seem having been processed with different tools. They should be homogeneised. Besides this, Figures 6 and Figure 8 and 9 should be moved forward, closer to their respective texts, because now they are behind the References section. Figures 8 and 9 are very small, as compared to the previous ones, and difficult to be read.

14. Are the number and quality of references appropriate? Yes, the list is quite generous and appropriate; references are cited timely in their correct place.

15. Is the amount and quality of supplementary material appropriate? Yes, as mentioned earlier, the authors make the C and Fortran versions of the code available upon request.

---

## Author Comment (AC1) · 22 Dec 2016

[10pt]article [authoryear,round]natbib [normalem]ulem color

We thank the reviewers for their careful inspection of the manuscript. In the following we address their comments point-by-point. We use *text in italics* to repeat the reviewer comments, normal text for our response, and **bold faced text** for quotations from the manuscript, with changes marked in colour.

We provide the revised manuscript (with and without changes highlighed) in the supplement.

[Figure]

**1 comments by Anonymous Referee #1**

1. *Title of the manuscript*
   *I think that the title of the manuscript does not fit to the content. From the title, I'm expecting a description of the retrieval system and a detailed analysis and evaluation of the products. Furthermore, I'm expecting a link where to receive the data or any other statement about how to receive the data. However, the manuscript mainly describes the retrieval system and some key properties of the retrieved products. The chapter on the product validation is rather a discussion on several possibilities how to validate the system or the products than an intense validation. Therefore I suggest to revise the title of the manuscript so that it rather gives a focus on the description of the processing system than the products.*

   We changed the title to read:

   **Consistent retrieval of land surface radiation products from EO Land Surface Products including Uncertainty Estimatestraceable uncertainty estimates**

   Regarding data and code availability we refer to the relevant section:

   **C and Fortran implementations of the two-stream code are available at http://fapar.jrc.ec.europa.eu. The JRC-TIP product based on MODIS collection 5 in 1 km resolution is available upon request to the authorscorresponding author.**

   Maybe the reviewer has over-looked this section, which (following the journal's guidelines) was placed after the references section. Or the reviewer found "authors" to general. We clarified.

2. *Writing style*
   *Many sentences are generally difficult to read because #1 they are very long; #2 they cover several topics, concepts, or ideas; or #3 the flow is interrupted*

[Figure]

*by insertions like "e.g.", "i.e", "for example", "however", or by references. Such sentences are for example on lines 27-30, 33-37, 37-39, 40-43, 44-46, 50-53, 63-67, 113-116, 215-217, and 341- 344. I suggest to shorten sentences, to remove insertions, and to place references at the end of sentences. Additionally, term like "for example" or "however" should be rather at the beginning of sentences than within sentences. The text needs to be revised in order to reduce the complexity of sentences and to make it more comprehensible.*

We made an attempt to follow the suggestion of the reviewer and simplified several long sentences in the manuscript.

We note, however, that we did not move the term "however" to the beginning of the sentence, because this would change the meaning of the word, see Strunk and White (1979), chapter IV: "Avoid starting a sentence with however when the meaning is "nevertheless." ... When however comes first, it means in whatever way or to whatever extent.". Also, where references were placed in the middle of a sentence this was mostly on purpose, to render the reference more precise. See next item for an example.

3. *Lines 28-30: The sentence is difficult to understand because the position of references split the flow. Please place all references at the end of the sentence.*

Here is the sentence the reviewer is referring to:

**The strength of this constraint is quantified by significant reductions of uncertainty in simulated terrestrial carbon fluxes diagnosed over (Kaminski et al., 2002; Rayner et al., 2005) or predicted after (Scholze et al., 2007; Rayner et al., 2011) the assimilation window.**

We deliberately put the references to Kaminski et al. (2002) and Rayner et al. (2005) after "fluxes diagnosed over", because they refer to studies, where fluxes are diagnosed over the assimilation window, whereas the references to Scholze

et al. (2007) and Rayner et al. (2011) are placed after "or predicted after", because these studies refer to prediced fluxes after the assimilation window.

4. *Lines 40-41: The text requires a high level of knowledge from the reader. For example, in this sentence the term "observation operator" is introduced just within two brackets. I think the text could be much more comprehensive if these kind of insertions are removed from sentences and are explained of defined afterwards in a second sentence.*

Good point, we made two sentences.

**The assimilation of an EO product such as FAPAR requires the capability to simulate (by a so-called *observation operator*) its counterpart . The task of an observation operator is to simulate the counterpart of an observation from the model's prognostic variables, i.e. the variables that the integration scheme of the model's dynamical equations steps forward in time (Kaminski and Mathieu, 2016).**

5. *Lines 93-96: This part would be easier to understand if you shortly define Lambertian scattering.*

We have added the definition:

**The model is designed to solve the radiation balance for the canopysoil system (see Figure 1). It simulates the solar radiant fluxes scattered by, transmitted through and absorbed in a vegetation canopy that is composed of so-called bi-Lambertian leaves (the radiation scattered from and transmitted through the leaves – featured as flat disks – does not have any angular dependency around the leaf normal vectors), possibly exhibiting a preferred orientation. The bi-Lambertian leaf scattering property is such that the fraction of radiation that is not absorbed is scattered as a cosine distribution around the leaf normal vectors.**

[Figure]

6. *Line 238: Use capital letters for the beginning of the sentence.*

   Thanks. We also removed a redundant "all":

   **shows all Figure 4 shows the location in the albedo plane of all albedo pairs ...**

7. *Lines 294-296: "is exhibits" is likely wrong.*

   Thanks, changed:

   **this reflects the influence of the soil background, which for LAI values in the range from 0.3 to 0.5 is exhibits an equally complex uncertainy structure**

8. *Line 297: "bottom right panel" = Fig. 5 f*

   Thanks, changed:

   **that is displayed in the bottom right panel panel f of Figure 5 we can clearly distinguish the two regimes.**

9. *Line 354-369: The purpose of this paragraph is unclear to me. Please provide first an introductory sentence that describes why the Beer-Bouger-Lambert law is introduced here and how it relates to the validation of the product.*

   We changed the text accordingly:

   **We note the one to one relation of the direct transmissionSuch measurements from LAI-2000 correspond to estimates of the fraction of the radiation that is transmitted through the vegetation canopy layers. When considering the direct radiation – which thus has not collided with the vegetation elements – the transmitted fraction, $T^{UnColl}$, and can be expressed with the classical Beer-Bouger-Lambert Law, where the exponential attenuation is a function of the effective LAI, $\tilde{LAI}$, through the Beer-Bouger-Lambert Law :**

$$T^{UnColl}(\mu_0) = exp(-\frac{1}{2}\frac{\tilde{LAI}(\mu_0)}{\mu_0}),\qquad(1)$$

where $\mu_0$ denotes the cosine of sun zenith angle (Pinty et al., 2006, 2009) , i.e. $\mu_0 = 1$ when the sun is at nadir. Figure 8 shows these observations together with the direct transmission derived by JRC-TIP from MODIS collection 5 broadband WSA products at 500m and 1km.

10. *Figure 7: Please explain shortly potential reasons for the differences because they seem to be associated with certain vegetation types, e.g. overestimation of "wrong" in transitional regions and underestimations in boreal forests.*

We extended the text as follows:

**The effect of changing the order of aggregation is quantified by Figure 7 which shows on global scale and for a particular Northern Winter period in 2005 the relative differences of FAPAR products derived by either aggregation order. Another point is that the aggregation also needs to be performed on the uncertainty. This requires a specification of spatial uncertainty correlation and is certainly less complicated at the albedo level than at the levelto The pattern of the differences suggests overestimation of FAPAR aggregated on the level of JRC-TIP products. products over boreal regions and an underestimation over transitional regions. One reason may be that running JRC-TIP on the aggregated albedo product does properly account for snow soil conditions at pixel level. Further analyses of underlying mechanisms are foreseen in a future study.**

11. *Table 1: Can you please indicate in the table legend or footnote why there are two mean values for the $\omega_l$ parameters.*

We extended the caption as follows:

[a] **Value associated with the 'green' leaf scenario, as opposed to the standard "polychromatic" leaf scenario.**

For further, related changes see next point.

[Figure]

12. *Figure 3: The figure would be much easier to understand if you could provide some more details on the shown data without forcing the reader to go to the original publication. What do the points represent? Are these results from measurements or model outputs? From which regions or ecosystems are the measurements?*

We extended the caption as follows:

**Prior 1.5 sigma uncertainty for effective canopy single scattering albedo (denoted by "leaves") and background reflectance (denoted by "snow" in case of snow and by "soils" otherwise). Mofified Modified version of Figure 1 from Pinty et al. (2008).**

And we also added clarifications to the discussion of the Figure:

**Figure 3 further shows the components of the prior PDF describing the effective canopy single scattering albedo, which is based on a combination of modelled and observed (Jacquemoud and Baret, 1990) and observed (Hosgood et al., 1995) leaf optical properties(see Pinty et al., 2008, for details). . They were further modified to best account for the overall effects on the domain-averaged radiant fluxes, of needle clumping into shoots, shoots or leaves clumping into crowns as well as the presence of woody elements in the canopy (see Pinty et al., 2008, for details). This ellipsoid describes the 1.5 uncertainty range of our standard prior, denoted as polyochromatic leaf scenario. We also operate the system with an alternative prior, denoted as green leaf scenario (not shown in figure 3), which is characterised by a much lower uncertainty and a slight shift in the mean value. Both scenarios are operated with snow and non-snow priors for the background reflectance, depending on additional snow information (both shown in figure 3).**

[Figure]

13. *Figures 4 and 5: The colour legends show that black is used for snow. However, I'm irritated from this legend because I don't see that black colour is used in any of these plots. Additionally, I think it is not necessary to indicate snow in these plots. Do I overlook something or could you remove the snow/black entry of the legend?*

Thanks. The snow indicator has been removed from the plots.

14. *Figure 8: The figure and the labels are too small. Please increase especially the size of labels.*

In both, Figure 8 and 9, the size of the labels (and the panels) are increased.

**2   comments by Anonymous Referee #2**

- *1. Does the paper address relevant scientific questions within the scope of BG? Yes, the paper shows the suitability of the previously developed JRC-TIP retrieval system to provide land surface parameters, including their respective full and rigorous uncertainty information, ready for assimilation into climate and numerical weather prediction models. The paper demonstrates this for LAI and FAPAR, also providing their lower resolution coarser aggregated products and validation performance, even with in situ dedicated instruments.*

  Thanks.

- *2. Does the paper present novel concepts, ideas, tools, or data? Yes, the paper shows the robustness and efficiency of the JRC-TIP package which could also be applied to new Copernicus products.*

  Thanks.

[Figure]

- *3. Are substantial conclusions reached? Yes, definitely: Flexibility of JRC-TIP to deliver radiatively consistent land surface products including their fluxes (absorbed, transmitted and reflected) suitable for assimilation into terrestrial models. Rigurous estimation and fully traceability of the uncertainty both for the state variables and for the simulated fluxes. Computationally efficient, robust and agile global-scale processing. Provision of aggregated coarser grids products better compliant with climate models. Possibility of various approaches to validation, including in-situ observations from dedicated instruments/sites*

  Thanks.

- *4. Are the scientific methods and assumptions valid and clearly outlined? Yes. It should be mentioned that the detailed scientific explanation of the inverse modelling approach requires high knowledgable proficiency on the matter. The excellent scientific basis of the senior co-authors guarantees this. They are absolute reference in the field.*

  Thanks.

- *5. Are the results sufficient to support the interpretations and conclusions? Yes. The work is presented for LAI and FAPAR. However, the title is ampler mentioning "Land Surface Products" in general. A minor suggestion is to better specify the title. The work has not been tested for many other different land surface products (soil moisture, evapotranspiration, etc).*

  Besides LAI and FAPAR, JRC-TIP retrieves a set of further land surface variables. In the manuscript we show, for example, single scattering albedo in the VIS (Figure 5), flux transmitted to the background in the VIS (Figure 8), background reflectance in the VIS and NIR.

  To make clear that the JRC-TIP does not directly retrieve, for example, evapotranspiration, we changed the title to read:

[Figure]

- *6. Is the description of experiments and calculations sufficiently complete and precise to allow their reproduction by fellow scientists (traceability of results)? Yes, but it is required that fellow scientists trying to reproduce the work should be proficient in the matter, as indicated in #4. Besides, the authors make the C and Fortran versions of the code available upon request.*

  Thanks.

- *7. Do the authors give proper credit to related work and clearly indicate their own new/original contribution? Yes. This is recognised in the generous and appropriate reference list, as well as in the Acknowledgment section.*

  Thanks.

- *8. Does the title clearly reflect the contents of the paper? The title is more general that the applications actually shown in the paper. As indicated in #5, the title mentions "Land Surface Products" in general. However, the application is only for LAI and FAPAR. It is suggested to be more specific in the title.*

  See response to #5.

- *9. Does the abstract provide a concise and complete summary? Yes, the abstract is very well and clearly written.*

  Thanks.

- *10. Is the overall presentation well structured and clear? Yes.*

  Thanks.

[Figure]

- *11. Is the language fluent and precise? Yes, as far as this referee may judge.*

  Thanks.

- *12. Are mathematical formulae, symbols, abbreviations, and units correctly defined and used? Yes.*

  Thanks.

- *13. Should any parts of the paper (text, formulae, figures, tables) be clarified, reduced, combined, or eliminated? A minor suggestion is to unify the format and style of the figures, namely Figure 3, Figures 4 and 5, Figure 6, and Figures 8 and 9. They are all graphs but seem having been processed with different tools. They should be homogeneised. Besides this, Figures 6 and Figure 8 and 9 should be moved forward, closer to their respective texts, because now they are behind the References section. Figures 8 and 9 are very small, as compared to the previous ones, and difficult to be read.*

  Thanks for the hints.

  We note that the graph type of Figure 3 (2 dimensional plane) is different from Figures 4 and 5 (2 dimensional plot, with third dimension indicated in colour), and Figures 8 and 9 have yet another graph type (time series).

  For homogenisation the following changes to the Figures were applied:

  Figure 6 uses the same sub-space and a colour code similar as Figure 5 b.

  In both, Figure 8 and 9, the size of the labels (and the panels) are increased.

  Regarding the position of the Figures, we'll see what can be done (together with the copy editor).

- *14. Are the number and quality of references appropriate? Yes, the list is quite generous and appropriate; references are cited timely in their correct place.*

  Thanks.

[Figure]

- *15. Is the amount and quality of supplementary material appropriate? Yes, as mentioned earlier, the authors make the C and Fortran versions of the code available upon request.*

  Thanks.

**References (in addition to those listed in the manuscript)**

Hosgood, B., S. Jacquemoud, G. Andre′oli, J. Verdebout, G. Pedrini, and G. Schmuck (1995), Leaf Optical Properties Experiment (LOPEX′ 93), Technical Report EUR 16095 EN, EC Joint Research Centre.

Jacquemoud, S., and F. Baret (1990), PROSPECT: A model of leaf optical properties spectra, Remote Sens. Environ., 34, 75– 91.

Strunk Jr W, White EB. The Elements of Style. Macmillan. New York. 1979.